# Livestock, pathogens, vectors, and their environment: A causal inference-based approach to estimating the pathway-specific effect of livestock on human African trypanosomiasis risk

**Julianne Meisner**[1,2]*, **Agapitus Kato**[3◉], **Marshall Lemerani**[4◉], **Erick Mwamba Miaka**[5◉], **Acaga Ismail Taban**[6◉], **Jonathan Wakefield**[7,8], **Ali Rowhani-Rahbar**[9], **David M. Pigott**[10], **Jonathan Mayer**[9], **Peter M. Rabinowitz**[1,2,11]

**1** Center for One Health Research, University of Washington, Seattle, Washington, United States of America, **2** Department of Global Health, University of Washington, Seattle, Washington, United States of America, **3** Uganda Virus Research Institute, Entebbe, Uganda, **4** Trypanosomiasis Control Program, Ministry of Health, Lilongwe, Malawi, **5** Programme National de Lutte contre la Trypanosomiase Humaine Africaine, Kinshasa, Democratic Republic of Congo, **6** HISP South Sudan, Juba, South Sudan, **7** Department of Biostatistics, University of Washington, Seattle, Washington, United States of America, **8** Department of Statistics, University of Washington, Seattle, Washington, United States of America, **9** Department of Epidemiology, University of Washington, Seattle, Washington, United States of America, **10** Department of Health Metrics Sciences, University of Washington, Seattle, Washington, United States of America, **11** Department of Environmental and Occupational Health Sciences, University of Washington, Seattle, Washington, United States of America

◉ These authors contributed equally to this work.
* meisnerj@uw.edu

**Data Availability Statement:** All data are fully available without restriction. HAT outcome data can

## Abstract

Livestock are important reservoirs for many zoonotic diseases, however the effects of livestock on human and environmental health extend well beyond direct disease transmission. In this retrospective ecological cohort study we use pre-existing data and the parametric g-formula, which imputes potential outcomes to quantify mediation, to estimate three hypothesized mechanisms by which livestock can influence human African trypanosomiasis (HAT) risk: the reservoir effect, where infected cattle and pigs are a source of infection to humans; the zooprophylactic effect, where preference for livestock hosts exhibited by the tsetse fly vector of HAT means that their presence protects humans from infection; and the environmental change effect, where livestock keeping activities modify the environment in such a way that habitat suitability for tsetse flies, and in turn human infection risk, is reduced. We conducted this study in four high burden countries: at the point level in Uganda, Malawi, and Democratic Republic of Congo (DRC), and at the county level in South Sudan. Our results indicate cattle and pigs play a reservoir role for the rhodesiense form (rHAT) in Uganda (rate ratio (RR) 1.68, 95% CI 0.84, 2.82 for cattle; RR 2.16, 95% CI 1.18, 3.05 for pigs), however zooprophylaxis outweighs this effect for rHAT in Malawi (RR 0.85, 95% CI 0.68, 1.00 for cattle, RR 0.38, 95% CI 0.21, 0.69 for pigs). For the gambiense form (gHAT) we found evidence that pigs may be a competent reservoir (RR 1.15, 95% CI 0.92, 1.72 in Uganda; RR 1.25,

be requested from the WHO (https://www.who.int/teams/control-of-neglected-tropical-diseases/human-africantrypanosomiasis/ research) and livestock density maps can be downloaded from https://github.com/JulianneMeisnerUW/LivestockMaps. Confounder and mediator data are available for download from the websites linked in our References. All analyses were performed in R, and all code are available in the same GitHub repository.

**Funding:** This work was supported by the National Institute of Environment and Health Science (NIEHS) (5T32ES015459-09 to JM). The funders had no role in study design, data collection and analysis, decision to publish, or preparation of the manuscript.

**Competing interests:** The authors have declared that no competing interests exist.

95% CI 1.11, 1.42 in DRC). Statistical significance was reached for rHAT in Malawi (pigs and cattle) and Uganda (pigs only) and for gHAT in DRC (pigs and cattle). We did not find compelling evidence of an environmental change effect (all effect sizes close to 1).

## Introduction

Domestic and wild animals—in particular bovids such as cattle, bushbuck, and African buffalo, and suids such as domestic pigs, bushpigs, and warthogs—are known to be important reservoirs for the acute form of human African trypanosomiasis, caused by *Trypanosoma brucei rhodesiense*(rHAT) [1–4], and there is some evidence that domestic pigs may also be reservoirs for the chronic form, caused by *T. b. gambiense* (gHAT) [5]. Hereafter, we use the term "livestock" to refer to cattle and domestic pigs.

However, livestock may also influence the distribution of HAT beyond their role as reservoir hosts. HAT is transmitted by tsetse flies, and several species which are important HAT vectors have been documented to prefer animal hosts, and to only bite humans during times of stress [6]. Also appreciated with the malaria vector *Anopheles arabiensis* [7–9], this preference may shield humans from tsetse bites and thus HAT exposure when favored animal hosts are nearby, termed zooprophylaxis. The fly biting behaviors that favor a zooprophylactic role versus a reservoir role are distinct, with strongly zoophilic tsetse being unlikely to be competent HAT vectors, and animal reservoirs unlikely to be important where the predominant HAT vector is strongly anthrophilic. Furthermore, zoonotic transmission requires host switching—whereby a given fly feeds on an animal host and, subsequently, a human host—while zooprophylaxis does not [10]. Tsetse preference for animal hosts is thus harmful (up to a threshold) for HAT control in the presence of an animal reservoir, and helpful in its absence. Consistent with this, gHAT models have found that in the absence of animal reservoirs tsetse biting preference for animals had a negative effect on HAT risk, while in the presence of animal reservoirs biting preference for humans became more influential [10]. More recently, mathematical modeling using data from DRC detected statistical support for animal reservoirs of gHAT, leading the authors to conclude that while animal reservoirs would be unlikely to maintain transmission on their own, they could stall progress towards elimination targets in some settings [11].

Livestock grazing can bring about ecosystem degradation, in the form of desertification (the conversion of fertile land to desert) in arid climates, increased woody plant cover in semi-arid rangelands, and deforestation in humid climates [12]. As early as 1988, it was noted that overgrazing could potentially change local and global weather systems [13]. Insect vectors of public health importance, including tsetse fly vectors of HAT, are sensitive to environmental factors [14]. In Burkina Faso and Ghana, cattle keeping, crop farming, and other climactic and anthropogenic factors have altered riverine landscapes, fragmenting gallery forests and influencing tsetse distribution [15]. Similarly, in Ethiopia, where animal African trypanosomiasis (AAT) is endemic but HAT is not, regulatory changes leading to expansion in agricultural land was documented to alter the distribution of tsetse flies [16]. Furthermore, the clearing of bush by subsistence livestock agriculture in pre-colonial Africa is credited with small-scale control of tsetse, termed "agricultural prophylaxis," and the resurgence of bush following catastrophic livestock losses to rinderpest ("cattle plague") is credited with the sleeping sickness outbreaks of the early 20th century [13].

Livestock may thus come to bear on HAT distribution through three hypothesized pathways: disease reservoir, zooprophylaxis, and environmental change. While One Health

approaches to HAT control target the reservoir pathway, the environmental change pathway was successfully employed for HAT and AAT control in pre-colonial Africa through agricultural prophylaxis, and in colonial Africa through environmentally-traumatic approaches such as brush clearing and widespread use of persistent organochlorine insecticides. Furthermore, vector control strategies that target livestock, such as insecticide treatment of cattle (ITC), exploit the close proximity of cattle to humans and their need to travel daily to the riverine habitats favored by gHAT vectors for water, and thus zooprophylaxis [17, 18].

Estimation of pathway-specific effects can narrow knowledge gaps surrounding the epidemiology of HAT, facilitate the targeted deployment of control measures, and provide key parameters for mathematical modeling. Previous authors have estimated elements of these effects in select foci, including the predictive ability of remote-sensed environmental data for tsetse distribution in Nuguruman, Kenya [19], and the effects of trypanocide mass-treatment of cattle on rHAT risk in Uganda [20], of habitat fragmentation on rHAT risk in Eastern Zambia [21], and of animal husbandry and climate on AAT risk in Burkina Faso [22]. Mediation analysis represents a powerful addition to this literature by quantifying the contribution of various mechanistic pathways to the total effect of livestock on HAT risk, allowing the estimation of the epidemiological significance of each pathway relative to the others. Such an approach is critical to addressing the significant uncertainty that remains surrounding the role of pigs as reservoirs of gHAT, and the ability of cattle to bring about environmental changes that in turn affect the distribution of HAT.

In this study, we build on our previous efforts to estimate the total effect of livestock density on HAT risk using the parametric g-formula [23], by applying the mediational g-formula to estimate the magnitude to which this effect is mediated by environmental change, in three high-burden countries: Malawi (rHAT), Uganda (rHAT and gHAT), and Democratic Republic of Congo (DRC, gHAT). We accomplished this using remote-sensed data, HAT surveillance data collated by the WHO Atlas of HAT, and our new time-series of high resolution livestock density maps [24]. We define livestock density as the ratio of livestock (separately for cattle and pigs) to humans, and fit all models separately for each country, for gHAT and rHAT, and for two mediators: normalized difference vegetation index (NDVI), a measure of vegetation cover, and land surface temperature (LST), both of which may be feasibly altered by livestock keeping or by the presence of livestock themselves. Additionally, in South Sudan (gHAT) we use a spatial extension of the regression approach to mediation analyses described by VanderWeele [25], and a single county-level map of livestock density produced for 2008 [24].

## Materials and methods

Our study is a retrospective ecological cohort study with cluster-year as the unit of analysis in Malawi, Uganda, and DRC, and county (administrative level 2) as the unit of analysis in South Sudan. This distinction was necessary as cluster-level geolocation was not available for livestock data in South Sudan [24]. Cluster is defined as a 0.017˚ pixel.

### Ethical approvals

As data collected were already de-identified, the University of Washington Institutional Review Board determined this work did not constitute engagement with human subjects (STUDY00004648). Similarly, no ethical review was required in DRC. Approvals were granted from the National Health Sciences Research Committee in Malawi (approval No. 2166), the South Sudan Research Ethics Committee (MOH/ERB21/2018), and the Uganda National Council of Science and Technology (ADM 194/212/01).

## Study population

In Uganda, DRC, and Malawi, our study population included all clusters within five hours' travel time of a fixed health facility capable of HAT diagnosis [26], defined separately for gHAT and rHAT. In South Sudan all counties which reported one or more HAT cases or conducted active surveillance during the study period comprised the study area.

The study period was defined separately for each country, on the basis of availability of HAT and livestock data: 2006–2018 for Uganda, 2003–2014 for Malawi, and 2010–2013 for DRC. In South Sudan the study period was restricted to 2008 alone as only the 2008 census provided geolocated (county-level) livestock ownership data for this country [24].

## DAG formulation

In all study countries we adopted the counterfactual approach to mediation analysis, identifying confounders by *a priori* subject knowledge, which we encoded in a directed acyclic graph (DAG; Fig 1). Index events are natural disasters or armed conflicts in both rHAT and gHAT models, and proximity to protected areas—a proxy for density of wildlife reservoirs of HAT—in rHAT models only. Time-varying variables are indexed by $t \in \{0, 1, \ldots T\}$, and variables in the minimally-sufficient adjustment set are boxed: {elevation, index events, LST, NDVI, wealth, vector control}.

Our DAG encodes our beliefs about the underlying temporal structure of these causal relationships. We believe LST, NDVI, and wealth exhibit exposure-confounder feedback, where LST, NDVI, and wealth at time *t* are a cause of livestock density at time *t*, which in turn is a cause of LST, NDVI and wealth at time *t* + 1. In the presence of exposure-confounder feedback confounding cannot be controlled using adjustment in regression models. Here we apply the mediational g-formula, one of Robin's generalized methods ("g-methods"), which allows estimation of causal effects in the face of time-varying confounding with exposure-confounder feedback [27–29]. As we did not have longitudinal data in South Sudan this issue did not arise; in this country, we applied the regression approach to mediation [27].

Note that while ethnicity, educational attainment, and religion are generally not time-varying at an individual-level, in our application these represent group-level variables (i.e., proportion receiving higher than a given level of education, proportion belonging to the majority ethnic group, and proportion belonging to Muslim, Christian, and other religions) and thus do vary in time.

The candidate confounders presented in this DAG do not represent a complete census of all factors that influence HAT distribution, as only variables which are either a cause of the exposure, or share a common cause with it, meet the necessary criteria to be a confounder [30]. For instance while medical interventions are critical determinants of HAT risk, they are not causes of livestock density, and their common causes are expected to be sufficiently approximated through wealth.

## Measures

**Exposure.**   Exposure is livestock density, parameterized as the ratio of livestock (cattle and pigs, separately) to humans using maps we detail in a separate publication [24]. In Malawi, Uganda, and DRC these were a time-series of continuous (raster) maps. Due to limited data availability, in South Sudan we produced a single areal (county-level) map for 2008.

**Outcome data.**   Outcome data included all new cases of HAT diagnosed in a given year and given cluster (Malawi, Uganda, DRC) or county (South Sudan) in the WHO Atlas of HAT [31]. Clusters which did not appear in the Atlas were either assigned 0 cases if they were at

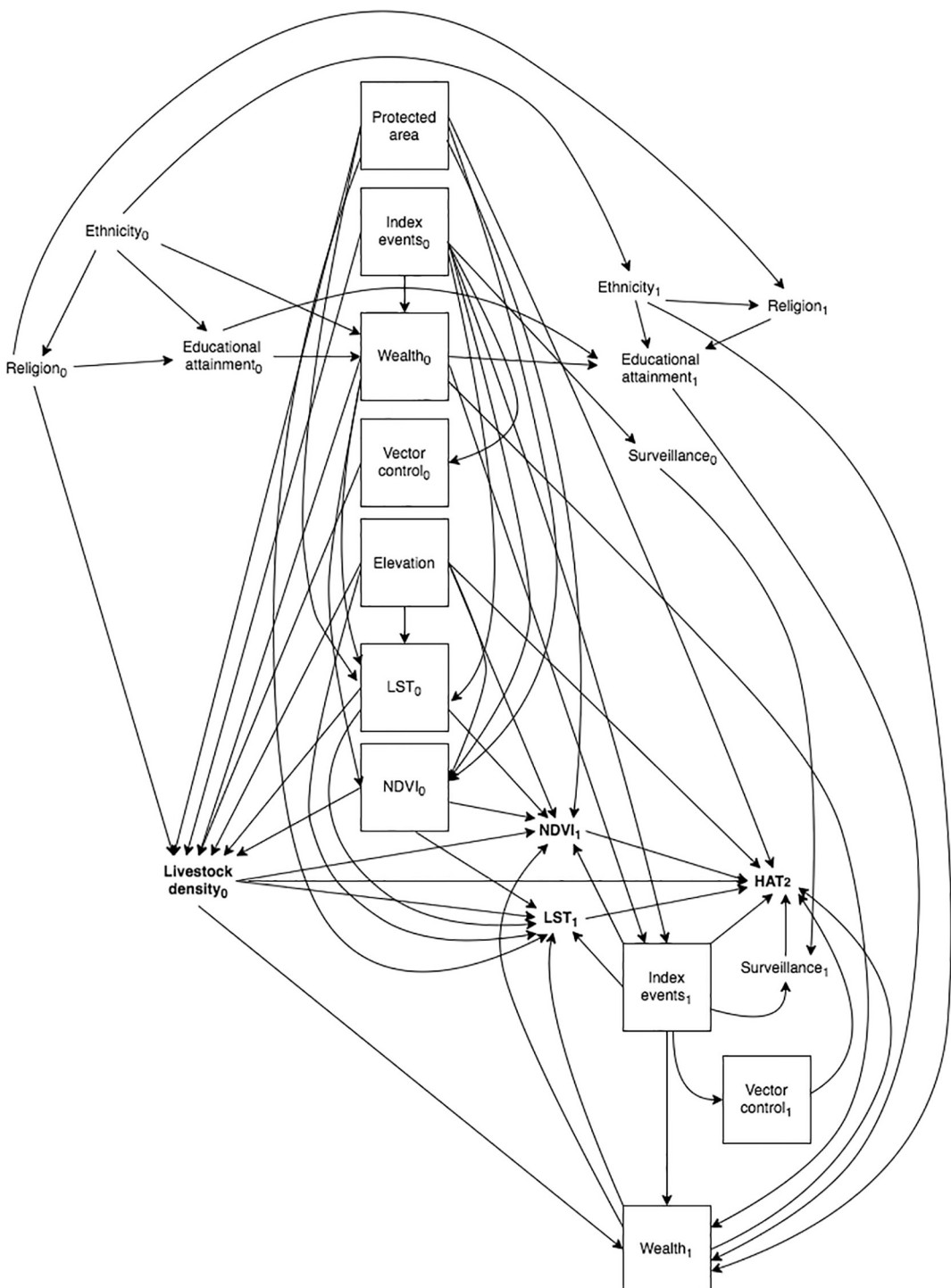

**Fig 1. Final time-varying directed acyclic graph (DAG), restricted to two time points.** Index events are natural disasters or armed conflicts. Protected areas pertains to rHAT models only. Exposure and outcome of interest are bolded, and confounders in the minimally-sufficient set are denoted by solid boxes.

least 1km from any clusters with reported cases, and removed from all analyses otherwise. Denominator data came from the University of Southampton's WorldPop project [32].

**Confounders.** Confounders are those in the minimum sufficient adjustment set, defined above (DAG formulation). We estimated wealth using an exploratory factor analysis approach modeled after the DHS Wealth Index [33] but excluding livestock-related measures. We then used spatial modeling to generate continuous annual maps in Uganda, Malawi, and DRC, and a single areal map for 2008 for South Sudan (detailed in S1 Appendix).

Natural disasters and armed conflicts were identified using the EM-DAT and UCDP/PIRO Armed Conflict Databases, respectively [34, 35]. NDVI data came from the Land Long Term Data Record 5 (LTDR5) [36]; LST data came from MODIS/Terra MOD21 for 2000–2002 [37], and from MODIS/Aqua MYD21 for 2003–2014 [38]. Finally, elevation data came from GMTED2010 [39], with the median product used.

We did not attempt to adjust for vector control as it is not practical to parameterize farmer-led efforts on the scale of this study, and top-down vector control efforts tend to be deployed across an entire epidemic focus. This means every location within an epidemic focus would have the same same value for every location (e.g., across the entire modeled Uganda gHAT focus [40]; Fig 2), and no adjustment can be performed.

## Causal estimands

The potential outcomes framework for causal inference defines causal effects in terms of counterfactual outcomes. Under this framework's cannonical notation, $Y$ refers to outcome (here, HAT), $M$ refers to mediator (here, NDVI and LST), $A$ refers to exposure (here, livestock density), and superscripts refer to counterfactual outcomes. Thus $Y^{aM^a}$ is the counterfactual outcome that would be observed with exposure fixed at value $a$ and the mediator taking the value that would be observed with exposure fixed to $a$ (note here that counterfactuals are defined both for $Y$ and for $M$, indicated by $M$ also carrying a superscript).

Mediation is defined through direct and indirect effects. Together, the indirect and direct effects comprise the total effect, which is the result one would get if no mediation analysis was conducted. The direct effect is the part of the total effect that is not mediated, while the indirect effect is the part of the total effect that "goes through" the mediator.

In the potential outcomes framework, estimands are the natural direct effect and the natural indirect effect. The word "natural" is used because the mediator is not fixed at set value $m$. Rather, the investigator fixes the exposure level at a set value $a$, and then the mediator takes the value it would naturally take under that exposure level. Thus, this mediator value is also a counterfactual, and the superscript notation is used: $M^a$.

To calculate each of these two estimands, two counterfactuals must be compared. For the natural direct effect, the comparison is between two counterfactuals in which exposure changes from $a^*$ to $a$, but the mediator does not change (it is $M^{a*}$ for both counterfactuals). For the natural indirect effect it is the opposite: the comparision is between two counterfactuals in which the exposure is fixed at $a$, but the mediator changes from $M^a$ to $M^{a*}$. In both cases, the estimand is represented as a population average effect: $E[Y^{aM^{a*}} - Y^{a*M^{a*}}]$ for the natural direct effect (mediator fixed, exposure changed) and $E[Y^{aM^a} - Y^{aM^{a*}}]$ for the natural indirect effect (mediator changed, exposure fixed).

In order for the population-average effect estimated from observed data to represent the true causal mediation effect, several criteria must be met, called the "identifiability criteria." These are (conditional) exchangeability, positivity, and stable unit treatment value assumption (SUTVA). SUTVA is comprised of consistency and no interference, detailed in S2 Appendix. Briefly, conditional exchangeability can be understood as no unmeasured confounding, and

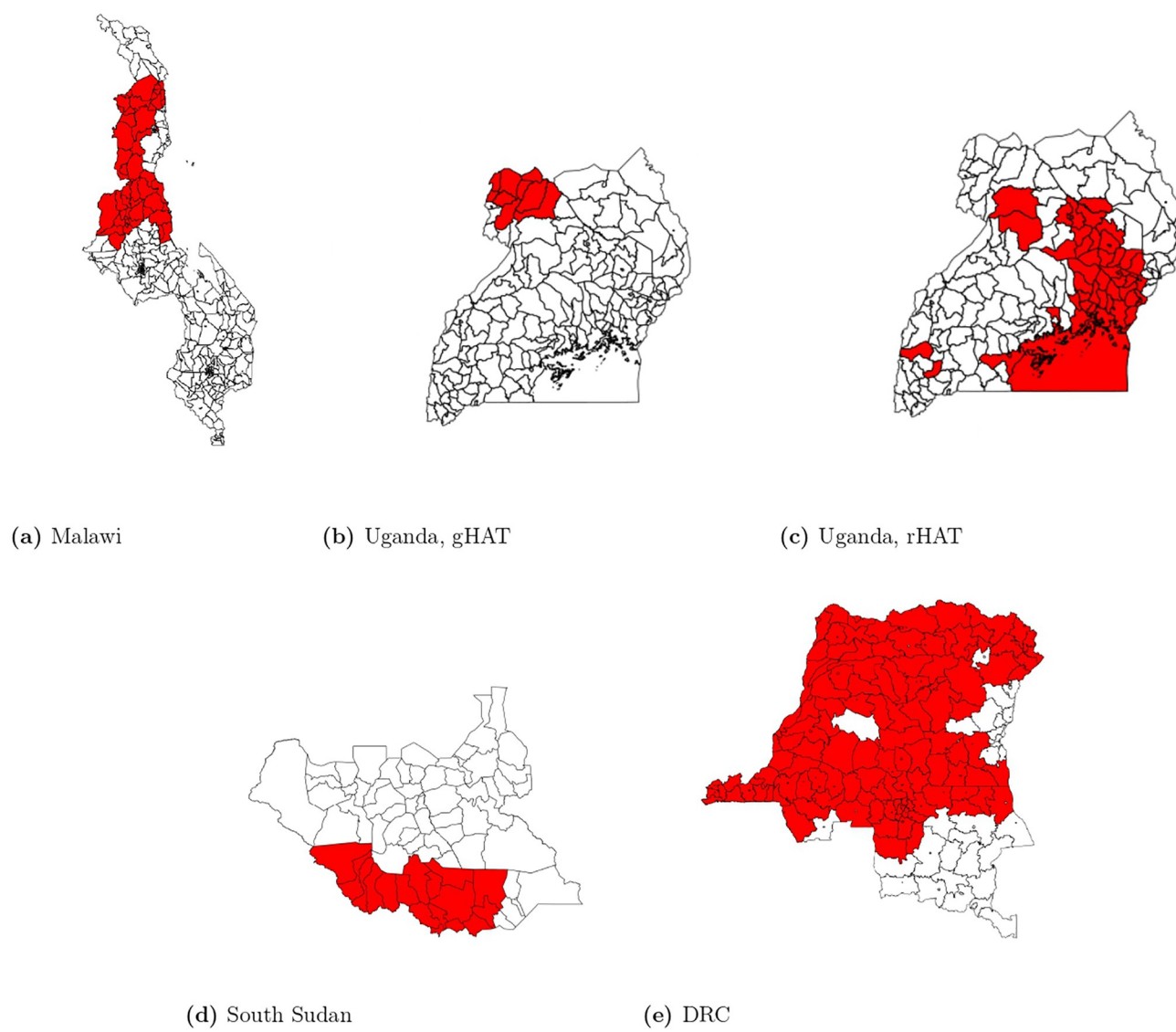

**(a)** Malawi    **(b)** Uganda, gHAT    **(c)** Uganda, rHAT

**(d)** South Sudan    **(e)** DRC

**Fig 2. National maps with administrative areas represented in the study data highlighted in red.** (a) Malawi: traditional authority (administrative level 3); (b-d) Uganda and South Sudan: county (administrative level 2); (e) DRC: territory (administrative level 2). All base maps were obtained from GADM (Malawi: https://geodata.ucdavis.edu/gadm/gadm4.1/shp/gadm41MWIshp.zip; Uganda: https://geodata.ucdavis.edu/gadm/gadm4.1/shp/gadm41UGAshp.zip; South Sudan: https://geodata.ucdavis.edu/gadm/gadm4.1/shp/gadm41SSDshp.zip; DRC: https://geodata.ucdavis.edu/gadm/gadm4.1/shp/gadm41CODshp.zip).

no interference can be understood as statistical independence between observations. Positivity and SUTVA have no corollaries outside of the potential outcomes framework, so we refer the reader to S2 Appendix for more detail.

The no interference assumption is likely violated in our study, the outcome is an infectious disease. We instead assume partial interference, in which each unit's potential outcome is independent of the rest conditional on a specified interference set $\chi_i = \{i_1, i_2, \ldots\}$ [41, 42]. Effectively, statistical independence can be assumed to hold outside of a buffer zone around each cluster. Reflecting the range of tsetse flies and grazing ranges of livestock, we define the interference set as all other clusters within a 5km radius [43, 44].

## Mediational g-formula

The mediational g-formula uses model-based standardization to derive these estimands; its implementation, along with the identifiability criteria, are detailed in S2 Appendix.

Briefly, the g-formula uses standardization to estimate counterfactual outcomes, which the parametric g-formula extends to settings with numerous or high-dimensional confounders using parametric models [29]. To implement the parametric g-formula, regression models (`glm` models in R, in this case) are fit separately for each time-varying confounder, mediator, and outcome, and for each year. Starting with the first year of the study, the investigator then uses the fitted models to predict the value of the time-varying confounder or outcome that would have been observed under the exposure value that they specify (i.e., if every cluster received exposure level $a$). These predictions then become inputs for prediction at the next time point, and so on through all of the years of the study. The DAG is used to encode each regression model: which predictors to include, appropriate lags, etc. This is conducted separately for NDVI and LST, and separately for each counterfactual which needs to be modeled to generate the desired causal estimands (natural direct and natural indirect effects, in this case). For instance, as the natural direct effect requires estimation of $Y^{aM^{a*}}$, we fixed exposure at $a$ when predicting out from the outcome and confounder models, and at $a^*$ when predicting out from the mediator models. The result is a distribution of outcome across all clusters and years. Uncertainty bounds are produced through non-parametric bootstrapping (100 simulations).

We defined $a^*$ as mean livestock density, and $a$ as mean × 1.5, and specify our causal estimand on the ratio scale (S2 Appendix), generating a rate ratio analog corresponding to a 50% increase in exposure. Our targets of inference are:

$$\text{rNDE} = E_{(s)}\left[E_t\left[E_i\left[Y^{\bar{a}M^{\bar{a}*}}_{t(s)i}/Y^{\bar{a}*M^{\bar{a}*}}_{t(s)i}\right]\right]\right]$$

$$\text{rNIE} = E_{(s)}\left[E_t\left[E_i\left[Y^{\bar{a}M^{\bar{a}}}_{t(s)i}/Y^{\bar{a}M^{\bar{a}*}}_{t(s)i}\right]\right]\right]$$

where $s$ indexes simulation (100 in total), $t$ indexes year, $i$ indexes cluster, and $Y^{\bar{a}M^{\bar{a}}}_{t(s)i}$ is the one-year cumulative incidence of HAT for a given livestock species, under $A = a$.

Validation of the mediational g-formula is conducted through implementation of the "natural course" model, whereby exposure (livestock density) is modeled as for the time-varying confounders. If the modeled distribution is close to the observed distribution, model performance is deemed adequate.

## Regression approach

Due to the absence of longitudinal exposure data in South Sudan, we applied a regression approach to our mediation analysis in this country. Under this approach, two models are fit—one for outcome, and one for mediator—and causal estimands are derived from linear combinations of parameters from these models.

Consistent with our DAG formulation, we assume that wealth measured in the 2008 South Sudan census is upstream of livestock density ascertained in the same census. Thus, wealth at time $t_1$ constitutes a mediator-outcome confounder which is downstream of exposure (as livestock density is a cause of wealth), precluding a simple regression approach to mediation analysis [27]. However, wealth was only measured at time 0 (2008), thus we proceeded with the regression approach and accept that our results will not have a causal interpretation.

The outcome (Poisson) model was as follows, where $c$ indexes county:

$$Y_c | \lambda_c \sim Poisson(P_c \lambda_c)$$

$$\lambda_c = exp\left(\theta_0 + \theta_1 \mu_c + \theta_2 m_c + \boldsymbol{\alpha}\mathbf{b}_c + S_c + \epsilon_c\right)$$

$$S_c | S_k, k \in ne(c) \sim N\left(\bar{S}_k, \frac{\sigma_s^2}{n_c}\right)$$

$$\epsilon_c | \sigma_\epsilon^2 \sim_{iid} N(0, \sigma_\epsilon^2)$$

where

- $Y_c$ is the number of cases in county $c$ in 2012

- $m_c$ is the mediator value in county $c$ in 2010, $m \in \{$NDVI, LST$\}$

- $\mu_c$ is estimated livestock density in county $c$ in 2008

- $\boldsymbol{\alpha}$ is a vector of coefficients

- $\mathbf{b}_c$ is a vector of exposure-outcome, exposure-mediator, and mediator-outcome confounders, $b \in \{$2008 wealth; 2007 NDVI, LST, conflicts, disasters; 2009 conflicts, disasters$\}$

- $S_c$ are county-level structured random effects which follow the ICAR model with marginal variance $\sigma_s^2$

- $ne(c)$ denotes neighbors (shared boundary) of county $c$

- $n_c$ is the number of neighbors of county $c$

- $\epsilon_c$ are unstructured (iid) county-level spatial random effects

- $P_c$ is the offset, given as population in county $c$

  and the mediator (linear) model was:

$$M_c | x_c \sim_{iid} N(x_c, \sigma_M^2)$$

$$x_c = \beta_0 + \beta_1 \mu_c + \boldsymbol{\gamma}\mathbf{b}_c + S_c + \epsilon_c$$

where

- $M_c$ is the mediator distribution in county $c$, $M \in \{$2010 NDVI, 2010 LST$\}$

- $\boldsymbol{\gamma}$ is a vector of coefficients

- $\mu_c$, $\mathbf{b}_c$, $S_c$, and $\epsilon_c$ are as defined above

  and ICAR is the intrinsic conditional autoregressive model, which smooths each county's random effect to that of its neighbors.

These models do not contain an exposure × mediator interaction term (imposing the assumption of linearity in the direct and indirect effects of exposure on outcome across levels of the mediator), thus the direct effect is then calculated as:

$$RR^{NDE} = exp[\theta_1 \times (\mu - \mu^*)]$$

and the indirect effect is calculated as:

$$RR^{NIE} = exp[\theta_2 \beta_1 \times (\mu - \mu^*)]$$

where $\mu - \mu^*$ are the two values of livestock density we are comparing [27]. We have parameterized exposure such that this contrast corresponds to a 50% increase in livestock density, allowing direct comparison with mediational g-formula results in the other countries. We used penalized complexity (PC) priors for the smoothing model, with $Pr(\sigma_\epsilon) > 1 = Pr(\sigma_s) > 1 = 0.01$. This yields a posterior 99% credible interval for each random effect's residual rate ratio of (0.36, 2.71) [45, 46].

We first fit naive models which did not account for measurement error in livestock density or wealth arising from the fact that these variables were estimated. We subsequently re-fit these specifying livestock density and wealth as random effects following a classical measurement error (MEC) model. We detail implementation of this model in S3 Appendix.

No off-the-shelf software is available to calculate standard errors for these estimands in the presence of spatial structure, thus we derived uncertainty estimates by taking 1,000 draws from the posterior distribution of each parameter in the above models, and taking the 2.5% and 97.5% percentiles of this distribution as the posterior 95% credible interval.

## Data and code availability

HAT outcome data can be requested from the WHO (https://www.who.int/trypanosomiasis_african/country/foci_AFRO/en/) and livestock density maps can be downloaded from GitHub https://github.com/JulianneMeisnerUW/LivestockMaps. Mediator and confounder data are available for download from the websites linked in our References. All analyses were performed in R, and all code are available in the same GitHub repository.

## Results

### Descriptive statistics

After removing clusters which did not meet the inclusion criteria, we were left with 3,982 clusters/year in Malawi, 5,746 clusters/year for gHAT models in Uganda, 13,570 clusters/year for rHAT models in Uganda, and 247,205 clusters/year in DRC. In South Sudan, the study area was comprised of 17 counties. General study areas are presented in Fig 2.

Malawi did not have any armed conflicts during the study period, and South Sudan did not have any natural disasters. Descriptive statistics are presented in Tables 1 and 2, with additional descriptive statistics figures presented in S4 Appendix.

### Mediational g-formula

Validation results indicated very strong agreement between the natural course model and reported HAT cases with the exception of 2010 and 2012 for the pig-gHAT models in Uganda. We removed these years from all subsequent analyses, meaning that they contributed to the models but not to the effect estimates presented herein as we do not trust their validity. Across the remaining years and models, the mean of the squared difference between observed livestock density and that predicted by the "natural course" model across cluster-years was $<1 \times 10^{-4}$.

Results are presented in Tables 3—5. In Uganda, Malawi, and DRC, we found no evidence of an indirect effect across both mediators, livestock species, and HAT types (all effect sizes close to 1).

**Table 1. Descriptive statistics, Malawi, Uganda, and DRC.**

| Variable | Mean (sd) | | | |
|---|---|---|---|---|
| | **Malawi** | **Uganda, gHAT** | **Uganda, rHAT** | **DRC** |
| Cattle density | 0.09 (0.31) | 0.31 (0.92) | 0.16 (0.42) | 0.06 (0.06) |
| Pig density | 0.09 (0.19) | 0.24 (0.72) | 0.06 (0.10) | 0.06 (0.03) |
| Elevation (meters) | 1,125 (280) | 890 (166) | 1,099 (79) | 524 (194) |
| HAT cases | 0.01 (0.14) | 0.04 (0.35) | 0.01 (0.15) | 0.02 (0.35) |
| LST (K) | 307 (4.7) | 281 (96) | 277 (95) | 306 (3.67) |
| NDVI | 0.15 (0.14) | 0.28 (0.13) | 0.26 (0.21) | 0.22 (0.18) |
| Wealth | 1.30 (1.04) | 0.84 (0.03) | 0.85 (0.03) | 0.25 (0.22) |
| Conflict | 0 (0%)* | 743 (68%)* | 1,824 (71%)* | 177,468 (9%)* |
| Disaster | | | | |
| Flood | 19,548 (33%)* | 413 (38%)* | 2,290 (0.9%)* | 46,921 (2.37%)* |
| Storm | 1,213 (2%)* | 0 (0%)* | 0 (0%)* | 0 (0%)* |
| Epidemic | 0 (0%)* | 2,065 (1.9%)* | 6,307 (2.5%)* | 28,343 (1.43%)* |
| Landslide | 0 (0%)* | 0 (0%)* | 127 (0.05%)* | 0 (0%)* |
| Drought | 0 (0%)* | 0 (0%)* | 8 (<0.01%)* | 0 (0%)* |
| Earthquake | 0 (0%)* | 0 (0%)* | 0 (0%)* | 410 (0.02%)* |
| Wildfire | 0 (0%)* | 0 (0%)* | 0 (0%)* | 16,575 (0.84%)* |

Descriptive statistics over study clusters and period (2000–2014 for Malawi, 2000–2018 for Uganda and DRC). Livestock density is parameterized as the ratio of animals to humans. sd: standard deviation.

*n(%)

In Uganda, a 50% increase in cattle density was associated with a 62% higher (95% CI 15% lower, 185% higher) risk of rHAT beyond the effect mediated by NDVI; a 68% higher (95% CI 16% lower, 182% higher) risk of rHAT beyond the effect mediated by LST; a 11% lower (95% CI 4% lower, 65% higher) risk of gHAT beyond the effect mediated by NDVI; and a 20% lower (95% CI 47% lower, 33% higher) risk of gHAT beyond the effect mediated by LST.

For pigs in Uganda, a 50% increase in density was associated with a 114% higher (95% CI 13% higher, 198% higher) risk of rHAT beyond the effect mediated by NDVI; a 116% higher

**Table 2. Descriptive statistics, South Sudan.**

| Variable | Mean (sd) |
|---|---|
| Cattle density | 0.77 (1.15) |
| Pig density | 0.02 (1.25) |
| Elevation (meters) | 693 (158) |
| HAT cases | 34 (52) |
| Number screened | 2,953 (4,200) |
| LST (K) | 312.54 (3.22) |
| NDVI | 0.28 (0.07) |
| Wealth | 0.19 (0.004) |
| Conflict | 8 (47%)* |

Descriptive statistics over study counties. 2008 data: cattle density (ratio of cattle to humans), pig density (ratio of pigs to humans), HAT cases, number screened, WorldPop population, LandScan population, wealth. 2007 data: LST, NDVI. 2006 data: conflicts. sd: standard deviation.

*n, (%)

**Table 3. Mediation analysis results, Uganda.**

| | | RR (95% CI) | |
|---|---|---|---|
| | Mediator | Direct effect | Indirect effect |
| | **Cattle** | | |
| rHAT | NDVI | 1.62 (0.85, 2.85) | 1.01 (0.88, 1.21) |
| | LST | 1.68 (0.84, 2.82) | 1.00 (0.87, 1.15) |
| gHAT | NDVI | 0.89 (0.56, 1.65) | 1.00 (0.89, 1.19) |
| | LST | 0.80 (0.53, 1.33) | 1.01 (0.90, 1.09) |
| | **Pigs** | | |
| rHAT | NDVI | 2.14 (1.13, 2.98) | 0.97 (0.87, 1.09) |
| | LST | 2.16 (1.18, 3.05) | 0.99 (0.89, 1.07) |
| gHAT | NDVI | 1.15 (0.92, 1.72) | 0.99 (0.91, 1.13) |
| | LST | 1.40 (0.85, 1.58) | 1.01 (0.94, 1.11) |

Mediational g-formula implemented such that effect estimates correspond to a 50% increase in livestock density. RR: rate ratio analog; 95% CI: credible interval over 100 iterations of the parametric g-formula

**Table 4. Mediation analysis results, Malawi.**

| | RR (95% CI) | |
|---|---|---|
| Mediator | Direct effect | Indirect effect |
| **Cattle** | | |
| NDVI | 0.85 (0.68, 1.00) | 1.03 (0.83, 1.31) |
| **Pigs** | | |
| NDVI | 0.38 (0.21, 0.69) | 1.01 (0.74, 1.27) |

Mediational g-formula implemented such that effect estimates correspond to a 50% increase in livestock density. LST not run for Malawi due to large amounts of missingness in the source data. RR: rate ratio analog; 95% CI: credible interval over 100 iterations of the parametric g-formula

(95% CI 18% higher, 205% higher) risk of rHAT beyond the effect mediated by NDVI; a 15% higher (95% CI 8% lower, 72% higher) risk of gHAT beyond that mediated by NDVI; and a 40% higher (95% CI 15% lower, 58% higher) risk of gHAT beyond that mediated by LST.

**Table 5. Mediation analysis results, DRC.**

| | RR (95% CI) | |
|---|---|---|
| Mediator | Direct effect | Indirect effect |
| **Cattle** | | |
| NDVI | 1.10 (1.00, 1.22) | 1.04 (0.97, 1.14) |
| LST | 1.18 (1.05, 1.30) | 0.99 (0.94, 1.06) |
| **Pigs** | | |
| NDVI | 1.21 (1.07, 1.33) | 1.02 (0.96, 1.08) |
| LST | 1.25 (1.11, 1.42) | 0.99 (0.95, 1.04) |

Mediational g-formula implemented such that effect estimates correspond to a 50% increase in livestock density. RR: rate ratio analog; 95% CI: credible interval over 100 iterations of the parametric g-formula

Thus results in Uganda were positive for pigs in both gHAT and rHAT foci, positive for cattle in rHAT foci, and negative for cattle in gHAT foci, reaching significance only for the pig-rHAT effect (Table 3).

In Malawi, we did not run LST models due to large amounts of missingness, and we removed protected area, elevation, and disaster from all models as very few HAT cases were observed during the study period leading to concerns regarding overfitting. For cattle, a 50% increase in density was associated with a 15% lower rHAT risk (95% CI 32% lower, null) beyond that mediated by NDVI. For pigs, a 50% increase in density was associated with a 62% lower (95% CI 79% lower, 31% lower) risk of rHAT, beyond that mediated by NDVI. Only pig results reached statistical significance (Table 4).

In DRC, a 50% increase in cattle density was associated with a 10% higher (95% CI 0% higher, 22% higher) risk of gHAT beyond the effect mediated by NDVI, and an 18% higher (95% CI 5% higher, 30% higher) risk of gHAT beyond the effect mediated by LST.

For pigs in DRC, a 50% increase in density was associated with a 21% higher (95% CI 7% higher, 33% higher) risk of gHAT beyond the effect mediated by NDVI, and a 25% higher (95% CI 11% higher, 42% higher) risk of gHAT beyond the effect mediated by LST.

DRC results are consistent with the gHAT results for pigs in Uganda, but discordant with cattle results; all DRC results reached statistical significance (Table 5).

## Regression

As the main models are expected to be biased, we will only interpret the MEC models here. A 50% increase in cattle density was associated with a 50% lower risk of gHAT (95% CI 91% lower, 54% higher) for the effect mediated by LST, however the effect mediated by NDVI was very close to 1, indicating no evidence of an indirect effect. Direct effects for cattle were also negative, with a 50% increase in density associated with a 78% lower (95% CI 87% lower, 55% lower) risk of gHAT beyond the effect mediated by NDVI, and a 50% increase in density associated with a 72% lower (95% CI 83% lower, 55% lower) risk of gHAT beyond the effect mediated by LST. Both direct effects reached significance, however neithter indirect effect was significant (Table 6).

For pigs, a 50% increase in density was associated with a 38% lower (95% CI 91% lower, 35% higher) risk of gHAT for the effect mediated by NDVI, and 43% lower (95% CI 87% lower, 115% higher) risk of gHAT for the effect mediated by LST. The direct effect not mediated by LST was similarly negative, with a 50% increase in density associated with a 14% lower (95% CI 56% lower, 68% higher) risk of gHAT, however the direct effect not mediated by NDVI was positive, with a 50% increase in density associated with a 55% higher risk of gHAT (95% CI 48% lower, 268% higher). All uncertainty bounds were wide for pigs, with none of these results reaching statistical significance (Table 6).

## Discussion

In Malawi, Uganda, and DRC, we did not find evidence that the effect of livestock on HAT risk is mediated by NDVI or LST. In South Sudan we did detect strong evidence of a mediation effect by LST for both cattle and pigs, and by NDVI for pigs. These results indicate the environmental pathway mediated by NDVI and LST is weak—if present—in rHAT and gHAT foci in Malawi, Uganda, and DRC. While point estimates were strong in South Sudan, uncertainty was generally large in this country, and these findings may furthermore be due to ecological bias (discussed in detail below).

For reservoirs, our direct (non-mediated effects) indicate that cattle and pigs may play a reservoir role for rHAT in Uganda and gHAT in DRC, and that pigs but not cattle may play a

**Table 6. Mediation analysis results, South Sudan.**

| Mediator | Model | RR (95% CI | |
|---|---|---|---|
| | | Direct effect | Indirect effect |
| Cattle | | | |
| NDVI | Naive | 0.34 (0.09, 1.28) | 0.99 (0.46, 1.60) |
| LST | Naive | 0.32 (0.12, 0.80) | 0.73 (0.09, 2.88) |
| NDVI | MEC | 0.22 (0.13, 0.41) | 0.98 (0.38, 1.55) |
| LST | MEC | 0.28 (0.17, 0.45) | 0.50 (0.09, 1.54) |
| Pigs | | | |
| NDVI | Naive | 0.96 (0.09, 8.85) | 0.75 (0.13, 3.59) |
| LST | Naive | 0.93 (0.26, 3.05) | 0.81 (0.13, 3.50) |
| NDVI | MEC | 1.55 (0.52, 3.68) | 0.62 (0.09, 1.35) |
| LST | MEC | 0.86 (0.44, 1.68) | 0.57 (0.13, 2.15) |

Estimates of direct (natural or controlled) effect and natural indirect effect, using 2012 HAT data as outcome, 2008 livestock density data as exposure, and 2010 NDVI and LST data as mediators, fit in two separate models. Adjustment performed for 2008 wealth, 2007 NDVI, 2007 LST, and 2007 and 2009 armed conflicts. Density is parameterized such that rate ratios correspond to a 50% increase in density. RR: rate ratio; CI: credible interval, taken over 1,000 draws from the posterior distribution of each parameter; Naive: model fit without accounting for measurement error in livestock density or wealth; MEC: measurement error model.

reservoir role for gHAT in Uganda. For zooprophylaxis, our results indicate pigs may play a zooprophylactic role for rHAT in Malawi, and cattle may play a zooprophylactic role for gHAT in Uganda. These results are largely consistent with our published total effect estimates [23], a finding that supports the validity of both sets of results as in the absence of an indirect effect, the direct effect is expected to resemble the total effect. In South Sudan we detected evidence of a zooprophylactic effect for cattle, but mixed evidence for pigs: the effect of pigs not mediated by NDVI was positive (reservoir), but the effect not mediated by LST was negative (zooprophylactic).

It is important to consider that these results are presented on the ratio scale, and thus should not be interpreted as a measure of impact. A risk factor for a disease with a very small (yet positive) effect size can still have substantial population-level impact if exposure to that risk factor is common, and conversely, a risk factor with a large effect size can have a very small population-level impact if exposure to that risk factor is very rare. Thus these results alone cannot tell us whether pigs or cattle play a large role in HAT transmission or maintenance, but rather whether there is evidence that such a role exists.

Where livestock play both a zooprophylactic and reservoir role, direct effect estimates reflect the balance between these opposing effects. For rHAT, our results provide evidence of a livestock reservoir effect in Uganda but not Malawi, where the zooprophylactic effect predominates. Our findings in Malawi are not necessarily discordant with what is known mechanistically—i.e., cattle and pigs are rHAT reservoirs—if this reservoir poses little risk to humans, allowing the zooprophylactic effect to dominate. This could arise if wildlife are the dominant source of human infection—due to host abundance, tsetse fly preference, contact patterns between hosts, adequate control of the domestic reservoir through insecticides or trypanocides, or other means. The spatial distribution of HAT in Malawi, where most active transmission occurs near protected areas, indicates this scenario may be consistent with HAT epidemiology in this setting [47]. A further mediation analysis examining an indicator of zooprophylaxis could be used to disentangle the zooprophylaxis and reservoir pathways, for

instance using an indicator of tsetse preference and host-tsetse contact. For gHAT, our results point to a reservoir role for pigs in both Uganda and DRC and provide some evidence that pigs are a reservoir in South Sudan. These results also provide equivocal evidence that cattle may serve as a gHAT reservoir: while there is evidence of a slight reservoir effect for gHAT in DRC, in Uganda and South Sudan a zooprophylactic effect was detected.

As with any imperfect study—interventional or observational, randomized or not—our results may also be the result of bias. Unmeasured confounding due to omission of vector control from our models, and the need to drop several confounders from our Malawi model due to overfitting concerns, may lead to residual confounding. As a result we encourage caution in interpreting the Malawi results. Vector control efforts could not be included in our model as we had no measure of farmer-led efforts, and top-down efforts did not have any variance within each epidemic focus modeled (i.e., would have the same value at every location and thus cannot be modeled and do not constitute confounders). Further, proximity to protected areas is an imperfect proxy for density of wildlife hosts, as wildlife can and do exist outside of protected areas.

Selection bias may arise if clusters or counties excluded from the study are actually at risk of reporting a HAT case, and differ systematically in their distribution of the exposure (livestock density), mediators (LST and NDVI) or confounders (index events, elevation, wealth, vector control). In South Sudan, this could arise if underreporting in the outcome data (discussed below) results in exclusion of an at-risk county from our analyses completely.

One of the major sources of bias in our study is underreporting in the outcome data. If such underreporting is non-differential with respect to HAT risk (i.e., clusters or counties at high risk of HAT are no more likely to underreport than those at low risk, even if high-risk individuals are less likely to enter the reporting system than low-risk individuals), effect estimates will be biased to the null in expectation. While weak surveillance infrastructure may drive an association between cluster- or county-level HAT risk and HAT reporting, we expect this mechanism to largely arise via civil unrest (conflict), which we have attempted to control for.

With the exception of South Sudan, we have not addressed uncertainty in livestock density and wealth in our estimates, and in South Sudan our implementation of the measurement error model assumes errors in wealth and livestock density are not correlated in space, which is likely violated. Measurement error may also arise in our confounders, driving residual confounding, and in our mediators (NDVI and LST), biasing direct and indirect effects.

In a country as large as DRC, there is likely subnational variation in the effect of livestock density on HAT risk. This could be captured by fitting models separately for subnational regions, for instance province. However given that all models need to be run four times (for all combinations of cattle, pigs, NDVI, and LST), this would markedly increase hypotheses tested. As a result, each country-level effect estimate we present here should be interpreted as a weighted average of locally-varying effects.

We have included conflict as a confounder since conflict can influence transmission of HAT through displacement and migration, and detection of HAT due to either weakened public health infrastructure, or enhanced surveillance by NGOs in migrant settlements. If these impacts are contained to the years in which conflicts were defined in our dataset, we do not expect our results to be biased. However the impacts of conflicts nearly always linger after the conflict, likely resulting in residual confounding during post-conflict years in the affected locations.

Finally, ecological bias arises from aggregation of variables, the presence of an unmeasured individual-level confounder whose association with exposure differs across groups, or the presence of an unmeasured individual-level effect modifier whose effect or distribution differs

across groups [48–50]. As ecological bias is sensitive to grouping definition, we are particularly concerned about its effects in South Sudan, where spatial units were large and irregular.

## Conclusion

Concerns about overfitting and residual confounding in the Malawi analyses, ecological bias in the South Sudan analyses, and underreporting in the outcome data for all countries aside, our study takes a novel approach to estimating the pathway-specific impact of cattle and pigs on gHAT and rHAT risk. By conducting our study across four high-burden countries, our findings lend strong supporting evidence in favor of a reservoir effect of pigs for gHAT. Our results also indicate that interventions on domestic animal reservoirs may be more likely to yield positive results in Uganda than Malawi, suggesting efforts should instead be targeted to vector control and strengthening passive surveillance systems. Control measures targeting the environmental change pathway are unlikely to be of much utility in these countries, their logistical and ethical challenges notwithstanding. Broad agreement with our previously-published total effect estimates [23] supports the validity of both results.

In conjunction with high-resolution livestock maps [24] these results can be interpreted on a policy-relevant scale, allowing policymakers and other stakeholders to identify priority areas for implementing domestic animal AAT control in a coordinated One Health framework.

## Supporting information

**S1 Appendix. Wealth mapping.** Methods and results.
(PDF)

**S2 Appendix. Motivation and implementation of the mediational g-formula.**
(PDF)

**S3 Appendix. Implementation of the measurement error model in South Sudan.**
(PDF)

**S4 Appendix. Descriptive statistics figures.**
(PDF)

## Acknowledgments

The authors wish to express sincere appreciation to the World Health Organization's Department of Control of Neglected Tropical Diseases for use of their Atlas of HAT data, and to the HAT research community, in particular Prof. Eric Fevré and Dr. Richard Selby.

## Author Contributions

**Conceptualization:** Julianne Meisner, Agapitus Kato, Marshall Lemerani, Erick Mwamba Miaka, Acaga Ismail Taban, Jonathan Wakefield, Ali Rowhani-Rahbar, Jonathan Mayer, Peter M. Rabinowitz.

**Data curation:** Julianne Meisner, Marshall Lemerani, Erick Mwamba Miaka, Acaga Ismail Taban.

**Formal analysis:** Julianne Meisner, Agapitus Kato, Erick Mwamba Miaka, Acaga Ismail Taban, Jonathan Wakefield, David M. Pigott.

**Funding acquisition:** Julianne Meisner.

**Investigation:** Julianne Meisner, Erick Mwamba Miaka, Ali Rowhani-Rahbar.

**Methodology:** Julianne Meisner, Marshall Lemerani, Acaga Ismail Taban, Jonathan Wakefield, Ali Rowhani-Rahbar, David M. Pigott, Jonathan Mayer, Peter M. Rabinowitz.

**Project administration:** Julianne Meisner, Agapitus Kato, Marshall Lemerani, Erick Mwamba Miaka, Peter M. Rabinowitz.

**Resources:** Julianne Meisner, Agapitus Kato, Peter M. Rabinowitz.

**Software:** Julianne Meisner, Jonathan Wakefield.

**Supervision:** Agapitus Kato, Marshall Lemerani, Erick Mwamba Miaka, Jonathan Wakefield, Ali Rowhani-Rahbar, David M. Pigott, Jonathan Mayer, Peter M. Rabinowitz.

**Validation:** Julianne Meisner, Jonathan Wakefield, Ali Rowhani-Rahbar, David M. Pigott.

**Visualization:** Julianne Meisner, Acaga Ismail Taban, Jonathan Wakefield, David M. Pigott, Jonathan Mayer, Peter M. Rabinowitz.

**Writing – original draft:** Julianne Meisner.

**Writing – review & editing:** Julianne Meisner, Agapitus Kato, Marshall Lemerani, Erick Mwamba Miaka, Acaga Ismail Taban, Jonathan Wakefield, Ali Rowhani-Rahbar, David M. Pigott, Jonathan Mayer, Peter M. Rabinowitz.

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
