## [Decision Letter · Decision Letter 0]

31 Aug 2023

PGPH-D-23-01458

Livestock, pathogens, vectors, and their environment: a causal inference-based approach to estimating the pathway-specific effect of livestock on human African trypanosomiasis risk

Dear Dr. Meisner,

Thank you for submitting your manuscript to PLOS Global Public Health. After careful consideration, we feel that it has merit but does not fully meet PLOS Global Public Health’s publication criteria as it currently stands. Therefore, we invite you to submit a revised version of the manuscript that addresses the points raised during the review process.

We look forward to receiving your revised manuscript.

Kind regards,

Najmul Haider, PhD

Academic Editor

Journal Requirements:

2. We ask that a manuscript source file is provided at Revision. Please upload your manuscript file as a .doc, .docx, .rtf or .tex.

3. Please provide separate figure files in .tif or .eps format only and remove any figures embedded in your manuscript file. Please also ensure all files are under our size limit of 10MB.

Additional Editor Comments (if provided):

Reviewers' comments:

Reviewer's Responses to Questions

**Comments to the Author**

1. Does this manuscript meet PLOS Global Public Health’s publication criteria? Is the manuscript technically sound, and do the data support the conclusions? The manuscript must describe methodologically and ethically rigorous research with conclusions that are appropriately drawn based on the data presented.

Reviewer #1: Yes

2. Has the statistical analysis been performed appropriately and rigorously?

Reviewer #1: Yes

3. Have the authors made all data underlying the findings in their manuscript fully available (please refer to the Data Availability Statement at the start of the manuscript PDF file)?

Reviewer #1: Yes

4. Is the manuscript presented in an intelligible fashion and written in standard English?

Reviewer #1: Yes

5. Review Comments to the Author

Reviewer #1: This is an excellent paper using a lot of ambition in the modelling approach. I have annotated very few comments on the PDF. You do not shy from the many limitations of such a study based on remote sensing data and remotely generated sources which have many potential weaknesses and biases. There is some lack of clarity on what you are talking about at times and of terms used which affects readers less familiar with your advanced modelling methodologies. Even use of species or common terminology for example on pigs - wild pigs domestic pigs? It is not clear always what you are talking about. There are also odd typographical errors mainly repetition of articles so the script needs careful screening by one or more authors. I feel there is sufficient evidence to support the general conclusions and it highlights the dangers of enabling domestic pig production in these countries which has many risks I believe as species acting as disease vessels and susceptible themselves to many prominent pathogens. High risk agricultural strategy and perhaps unwarranted on this continent. The low levels of HAT in many pastoral systems - see recent papers on participatory epidemiology on this subject from the region e.g. northern Kenya - which suggest as you do that some factor such as zooprophylaxis is at work indeed amongst cattle keepers. This is in conflict with prioritisation exercises on zoonosis in this region from higher authorities so this modelling will perhaps help clarify matters and reduce assumptions for better policies. Minor effort should see this through.

6. PLOS authors have the option to publish the peer review history of their article (what does this mean?). If published, this will include your full peer review and any attached files.

**Do you want your identity to be public for this peer review?** For information about this choice, including consent withdrawal, please see our Privacy Policy.

Reviewer #1: **Yes: **Richard Kock

---

## [Decision Letter · Decision Letter 1]

10 Oct 2023

Livestock, pathogens, vectors, and their environment: a causal inference-based approach to estimating the pathway-specific effect of livestock on human African trypanosomiasis risk

PGPH-D-23-01458R1

Dear Dr. Meisner,

We are pleased to inform you that your manuscript 'Livestock, pathogens, vectors, and their environment: a causal inference-based approach to estimating the pathway-specific effect of livestock on human African trypanosomiasis risk' has been provisionally accepted for publication in PLOS Global Public Health.

Best regards,

Najmul Haider, PhD

Academic Editor

Reviewer Comments (if any, and for reference):

Reviewer's Responses to Questions

**Comments to the Author**

1. If the authors have adequately addressed your comments raised in a previous round of review and you feel that this manuscript is now acceptable for publication, you may indicate that here to bypass the “Comments to the Author” section, enter your conflict of interest statement in the “Confidential to Editor” section, and submit your "Accept" recommendation.

Reviewer #1: All comments have been addressed

2. Does this manuscript meet PLOS Global Public Health’s publication criteria? Is the manuscript technically sound, and do the data support the conclusions? The manuscript must describe methodologically and ethically rigorous research with conclusions that are appropriately drawn based on the data presented.

Reviewer #1: Yes

3. Has the statistical analysis been performed appropriately and rigorously?

Reviewer #1: Yes

4. Have the authors made all data underlying the findings in their manuscript fully available (please refer to the Data Availability Statement at the start of the manuscript PDF file)?

Reviewer #1: Yes

5. Is the manuscript presented in an intelligible fashion and written in standard English?

Reviewer #1: (No Response)

6. Review Comments to the Author

Reviewer #1: I am satisfied with teh revision

7. PLOS authors have the option to publish the peer review history of their article (what does this mean?). If published, this will include your full peer review and any attached files.

**Do you want your identity to be public for this peer review?** For information about this choice, including consent withdrawal, please see our Privacy Policy.

Reviewer #1: **Yes: **Richard Anthony Kock
